# High-Level Production of a Thermostable Mutant of *Yarrowia lipolytica* Lipase 2 in *Pichia pastoris*

**DOI:** 10.3390/ijms21010279

**Published:** 2019-12-31

**Authors:** Qinghua Zhou, Zhixin Su, Liangcheng Jiao, Yao Wang, Kaixin Yang, Wenjuan Li, Yunjun Yan

**Affiliations:** Key Laboratory of Molecular Biophysics, the Ministry of Education; College of Life Science and Technology, Huazhong University of Science and Technology, Wuhan 430074, China; qinghuazhou1@126.com (Q.Z.); xuzhixin5512@163.com (Z.S.); jiaoliangcheng@gmail.com (L.J.); wangyao_1962@163.com (Y.W.); ykxllq@163.com (K.Y.); 18238636670@139.com (W.L.)

**Keywords:** *Yarrowia lipolytica* lipase 2, thermostability, *Pichia pastoris*, heterologous overexpression, fermentation

## Abstract

As a promising biocatalyst, *Yarrowia lipolytica* lipase 2 (YlLip2) is limited in its industrial applications due to its low thermostability. In this study, a thermostable YlLip2 mutant was overexpressed in *Pichia pastoris* and its half-life time was over 30 min at 80 °C. To obtain a higher protein secretion level, the gene dosage of the mutated *lip2* gene was optimized and the lipase activity was improved by about 89%. Then, the YlLip2 activity of the obtained strain further increased from 482 to 1465 U/mL via optimizing the shaking flask culture conditions. Subsequently, Hac1p and *Vitreoscilla* hemoglobin (VHb) were coexpressed with the YlLip2 mutant to reduce the endoplasmic reticulum stress and enhance the oxygen uptake efficiency in the recombinant strains, respectively. Furthermore, high-density fermentations were performed in a 3 L bioreactor and the production of the YlLip2 mutant reached 9080 U/mL. The results demonstrated that the expression level of the thermostable YlLip2 mutant was predominantly enhanced via the combination of these strategies in *P. pastoris*, which forms a consolidated basis for its large-scale production and future industrial applications.

## 1. Introduction

The microbial lipases are extremely important biocatalysts in the industrial field owing to their characteristics of high catalytic activity, mild reaction conditions, and rapid reaction [1,2,3,4]. As a typical lipase, *Yarrowia lipolytica* lipase 2 (YlLip2) has been extensively applied in food processing, biodiesel production, and resolution of chiral compounds [5,6,7]. The YlLip2 activity was reported to be 12,500 U/mL in a 7.5 L bioreactor in *Pichia pastoris* X33 [8] and 42,900 U/mL in 5 L fed-batch fermentation in *P. pastoris* GS115 [9]. However, wild-type YlLip2 is inactivated very quickly at high temperatures, and its half-life time is only about 1.5 min at 60 °C [5,7]. Thus, the application of YlLip2 has been severely hampered due to its low temperature tolerance. Researchers have been devoted to improving the thermostability of YlLip2 [10,11,12], and a breakthrough indicated that the half-life time of YlLip2 at 60 °C could increase 127-fold if the 244 Cys was replaced with an Ala [12]. Nevertheless, the reported YlLip2 mutant exhibited very low lipase activity (only 2.02 U/mL) in *Y. lipolytica*, which was 12% lower than the wild-type YlLip2 observed in a shaking flask (2.3 U/mL). This still hinders the large-scale industrial application of the YlLip2 mutant. Therefore, it is urgent to develop strategies to significantly improve the expression level of the thermostable YlLip2 mutant.

The methylotrophic yeast *P. pastoris* has been deeply studied, mainly due to its merits: it can realize high-density cultivation; there are various expression vectors and strains for choice; it can effectively reduce endotoxin and phage pollution; the target gene can be easily integrated into the genome; the downstream separation and purification of the target protein are very convenient [13,14,15,16]. As is known, the expression level of heterologous proteins in *P. pastoris* is affected by many factors, such as codon usage bias, promoter strength, gene dosage, and the protein translocation process [17,18,19]. Among these, gene dosage is one of the most important factors, and increasing the gene copy number may significantly improve the expression level of heterologous proteins [18,19]. Recently, Jiao et al. [20] obtained a recombinant strain with five copies of the *Rhizopus oryzae* lipase gene (*rol*) and its lipase activity was eightfold higher than that of the strain carrying a single copy of the *rol* gene. On the other hand, some studies have conversely shown that a high gene copy number may decrease the secretion level of heterologous proteins [20,21]. Part of the reason is that when heterologous genes are overexpressed, large amounts of new peptide chains may fail to process or fold correctly, and the misfolded proteins cannot be timely secreted out of cells, which leads to the unfolded protein response (UPR) [21,22]. Then, some UPR signals will activate the splicing of *hac1* mRNA and the spliced mRNA can be translated to yeast transcription factor Hac1p [23,24]. Hac1p can regulate the transcription level of downstream genes in UPR and enhance the capacity of the endoplasmic reticulum (ER) for protein folding and modification [25,26]. Studies have indicated that coexpression of Hac1p could improve the transportation and secretion of heterologous proteins in *P. pastoris* [21,24,27]. However, till now, there has been no research on the coexpression of Hac1p and YlLip2.

The recombination strains themselves determine the expression level of heterologous proteins, and excellent fermentation conditions are the premise for the full expression of target genes [28,29,30]. As such, dissolved oxygen (DO) is a crucial parameter and often becomes a limitation for industrial fermentation production in *P. pastoris* [28,29]. The *Vitreoscilla* hemoglobin (VHb) from *Vitreoscilla stercoraria*, a terminal oxidase, can facilitate oxygen delivery and improve respiration metabolism [31]. Wang et al. [32] enhanced the expression level of YlLip2 to 1.31 times compared with the control strains via coexpressing the *vgb* and *lip2* genes. As a consequence, coexpression of VHb is an effective way to alleviate the oxygen limitation during high-density fermentation in *P. pastoris* [31,32].

Therefore, in this study, a thermostable YlLip2 mutant was obtained by substituting the 244 Cys with an Ala. Subsequently, expression vectors carrying different copies of the mutational *lip2* gene were constructed in vitro for optimization of the gene dosage, and then fermentation conditions were further optimized for increasing YlLip2 production. In addition, the *hac1* and *vgb* genes were respectively coexpressed with the *lip2* gene to mitigate UPR and oxygen limitation in cells. Based on this, the sieved recombinant strains were fermented in a 3 L bioreactor to achieve the maximal lipase secretion level. The results revealed that the YlLip2 mutant expression level was predominantly improved by the strategies employed in this study, which demonstrates the possibility of their convenient use in subsequent industrial applications.

## 2. Results 

### 2.1. Lipase Activity Analysis for Transformants with Different Copies of the lip2 Gene

The multicopy plasmids were assembled and validated to determine the optimal gene dose for YlLip2 expression in *P*. *pastoris* (Appendix A). There were two *Bgl*II and one *Bam*HI restriction enzyme cutting sites on the expression vectors pAOα-nlip2, and after double digestion with *Bgl*II/*Bam*HI, the bands 2403, 4027, and 2575 × n bp (*n* = 1–4) were clearly visible in agarose gel electrophoresis (Appendix A). Recombinant strains GS115/pAOα-nlip2 (*n* = 1–4) harboring different *lip2* gene copies were incubated in 500 mL shaking flasks under the previously stated procedures to screen the optimal gene dosage for YlLip2 expression in *P*. *pastoris*. As shown in Figure 1, the colony GS115/pAOα-2lip2 #15 attained the highest lipase activity at 482 U/mL and was about 89% higher than GS115/pAOα-lip2 strains (255 U/mL). Four transformants with higher lipase activity were chosen as representatives of GS115/pAOα-nlip2 (*n* = 1–4) recombinant strains, and the *lip2* gene copy number was determined via quantitative PCR (qPCR). As a result, all transformants harbored the exact same *lip2* gene copy numbers as the corresponding plasmids did (Table 1). 

### 2.2. Characterization Analysis of the YlLip2 Mutant

In this study, the biochemical characterization of the YlLip2 mutant produced by *P*. *pastoris* was conducted first. As shown in Figure 2A,B, the optimum pH for the YlLip2 mutant was pH 8.0 and the optimum temperature was 35 °C; these were consistent with previous reports on YlLip2 [10,11,12]. Furthermore, the relative activity of the mutational YlLip2 after incubation at different temperatures was determined (Figure 2C). In comparison with the low thermostability of the wild-type YlLip2, the half-life time of the YlLip2 mutant was over 30 min at 80 °C, which was convenient for industrial applications.

### 2.3. Shaking Flask Culture Optimization for YlLip2 Mutant Production

As is known, heterologous protein production in recombinant strains is affected by many culture parameters [12,20]. Thus, the shaking flask culture parameters were optimized successively to screen subsequent transformants and enhance YILip2 mutant production in the recombinant strains (Appendix A). Under the optimized culture parameters (initial pH 7.0, addition of 1% (*v*/*v*) absolute methanol daily, initial growth medium volume of 30 mL, inoculation density of 3% (*v/v*), and fermentation for 144 h at 22 °C), the transformant GS115/pAOα-2lip2 #15 reached the highest lipase activity at 1465 U/mL, which was increased about twofold compared with the original culture parameters. Therefore, optimizing culture conditions was of great significance to improve the secretion level of heterologous proteins in *P. pastoris*.

### 2.4. Fed-Batch Fermentation, SDS-PAGE, and Mass Spectrometry

The fermentation of the transformant GS115/pAOα-2lip2 #15 was performed in a 3 L bioreactor. The result suggested that the lipase activity, dry cell weight, optical density at 600 nm (OD_600_), and total protein concentration continuously rose to the maximum value at 142–160 h (Figure 3A). Compared with the shaking flask culture, high-density fermentation significantly improved the expression level of the YILlip2 mutant in GS115/pAOα-2lip2 #15 (from 1465 to 6300 U/mL). 

The supernatants from the 3 L bioreactor fermentation of GS115/pAOα-2lip2 #15 were mixed with 2× SDS-PAGE loading buffer (Takara Biotechnology Co., Ltd, Dalian, China) and run on SDS-PAGE gels. A protein band corresponding to YlLip2 (~37 kDa) was clearly present (Figure 3B) and thickened with continuous fermentation. It could also be discerned that the YlLip2 mutant was the main extracellular product of the recombinant strains, which is beneficial for simplifying the downstream purification process in large-scale production. Furthermore, the target protein band was identified by mass spectrometry, and the results indicated that most peptides successfully matched with YlLip2 (Appendix A).

### 2.5. Effect of Coexpression Hac1p on YlLip2 Secretion

In the abovementioned results, the optimal copy number of the mutated *lip2* gene was lower than in previous experiments [9,32]; hence, we suspected that it would be more difficult to fold and translate the YlLip2 mutant in cells. Thus, the pPIC3.5k-hac1 vector was used to transform GS115/pAOα-2lip2 #15 to obtain the recombinant strains GS115/2lip2-3.5Khac1. As shown in Appendix A, the lipase activity of 15 colonies was measured under the optimized culture conditions in shaking flasks and the transformant GS115/2lip2-3.5Khac1 #32 obtained higher lipase activity at 2200 U/mL, which was a 50% improvement compared with GS115/pAOα-2lip2 #15. Moreover, the lipase activity of GS115/2lip2-3.5Khac1 #32 in the 3 L bioreactor fermentation reached 8150 U/mL and was 29% higher than that observed in GS115/pAOα-2lip2 #15 (Figure 4). The qPCR analysis displayed that the GS115/2lip2-3.5Khac1 #32 harbored two copies of the *hac1* gene, indicating a single copy of an additional *hac1* gene was inserted into the recombinant strain genome (Table 1). 

### 2.6. Influence of Coexpressing vgb on YlLip2 Production

pPICZA-vgb was introduced into GS115/pAOα-2lip2 #15 to explore the effect of coexpressing VHb on YlLip2 production. Nineteen colonies of the recombinant strains GS115/2lip2-ZAvgb were incubated under the optimized culture parameters in shaking flasks, and the highest lipase activity from GS115/2lip2-ZAvgb #13 achieved 2060 U/mL (Appendix A). CO-difference spectrophotometric analysis was used to confirm the functional production of VHb in the GS115/2lip2-ZAvgb #13 strain. As shown in Appendix A, there was a characteristic absorption peak of CO binding at 420 nm in the spectrum of the VHb^+^ strain GS115/2lip2-ZAvgb #13, whereas no such peak was detectable in the VHb^−^ strain GS115/pAOα-2lip2 #15. The *vgb* gene copy number of GS115/2lip2-ZAvgb #13 was determined via qPCR and the result showed that there was only one copy of the *vgb* gene in the GS115/2lip2-ZAvgb #13 genome (Table 1). Furthermore, the VHb^+^ and VHb^−^ strains were incubated in 3 L bioreactor fermentation with higher and lower DO levels (about 50% and 10%). The lipase activity of the VHb^+^ strain reached 7650 U/mL and the total protein content was 5.1 g/L after 115 h under the higher DO concentration, which was about 21% higher than that of the VHb^−^ strain GS115/pAOα-2lip2 #15 (Figure 5A). Under the lower DO level, the highest YlLip2 activity of the VHb^+^ strain was 6950 U/mL, but the activity of the VHb^−^ strain was only 3700 U/mL (Figure 5B). It is also worth pointing out that there were no marked differences for the growth status between the VHb^+^ and VHb^−^ strains, confirming that the expression of VHb did not affect cell growth.

### 2.7. Coexpression of Hac1p and VHb to Increase YlLip2 Expression Level

Respective coexpression of Hac1p or VHb could improve YlLip2 production in *P. pastoris*. Thus, aiming at further increasing the level of expression of the mutant lipase, we coexpressed both Hac1p and VHb with YlLip2. The plasmid pPICZA-vgb was used to transform GS115/2lip2-3.5Khac1 #32, and 19 colonies of recombinant strains GS115/2lip2-hac1-vgb were incubated in shaking flasks. As shown in Appendix A, the transformant GS115/2lip2-hac1-vgb #21 attained the highest lipase activity at 2630 U/mL. Moreover, the 3 L bioreactor fermentation for GS115/2lip2-hac1-vgb #21 obtained the maximal lipase activity at 9080 U/mL, which was increased by 11.4% and 18.7% compared with GS115/2lip2-3.5Khac1 #32 and GS115/2lip2-ZAvgb #13, respectively (Figure 6). Similarly, it was found that a single copy of the *vgb* gene was inserted into the recombinant strain GS115/2lip2-hac1-vgb #21 genome by qPCR analysis. In previous studies, some YlLip2 mutants showed excellent thermostability but very low lipase activity in yeast species (Table 2). However, the YlLip2 mutant in this study obtained a very high expression level and exhibited superior thermostability, which deserves further research and offers a promising perspective for future industrial-scale production.

## 3. Discussion

In the present study, an efficient method was used to explore the effect of gene dosage on YlLip2 expression in *P. pastoris*. The recombinant strains carrying the optimal gene copy number could be screened from only 32 colonies. The exact target gene copies in different transformants could be easily predicted without determination by qPCR because all transformants carried the same *lip2* gene copies as the corresponding plasmids did. Compared with traditional strategies which often increase the copy number of target genes in the *P. pastoris* genome via post-transformational vector amplification [33], the method here is more controllable and accurate.

Besides gene dosage, the expression level of YlLip2 was also successfully enhanced by optimizing the shaking flask culture parameters. This result is consistent with many other studies on shaking flask culture optimization for improving protein expression in *P*. *pastoris* (Appendix A) [20,33,34,35,36,37]. However, the effects of culture parameters on the host strain and the expressed protein may not always be the same. For example, a high temperature is beneficial for cell growth but may not be for protein production because a high temperature inhibits the activity of the AOX1 promoter [38]. In near-neutral pH medium, recombinant strains grow well and more target proteins would be secreted out of the cells [39]. Methanol is often utilized as a carbon source in shaking flask fermentation, but *P*. *pastoris* cannot tolerate higher concentrations of methanol, so the methanol addition concentration is about 1% (*v/v*) daily [40]. A relatively small culture medium volume in a shaking flask is convenient for oxygen delivery but not for cell growth, owing to the scarcity of nutrition [41]. The incubation time and inoculation density usually depend on host strain and target protein [17]. Above all, for each obtained engineered strain, researchers optimize its shaking flask culture parameters for a better understanding of its full potential for target protein production.

Usually, overexpression of heterologous protein in recombinant strains may cause improper protein folding and lead to UPR [22,23]. The underlying reason is that some expression-related helper proteins become limitation factors during protein synthesis, such as Ssa4, Bmh2, Pdi, and Bip [21], while Hac1p can activate the transcription and translation of these helper proteins [24,25]. On the basis of previous studies [21,22,23,24,25], we supposed that one copy of the native *hac1* gene was insufficient to reduce the ER stress caused by the overexpression of the YlLip2 mutant, and coexpression of Hac1p could effectively enhance the ability of ER to process and fold peptide chains. Thus, the Hac1p was further coexpressed with the YlLip2 mutant. The result confirms our supposition and indicates a great improvement in the production of the YlLip2 mutant in *P. pastoris*.

DO is a crucial parameter for the high-density fermentation of *P*. *pastoris* [28]. Previous studies revealed that the metabolic pathways of *P. pastoris* would change to produce ethanol and subsequently inhibit protein synthesis when yeast cells were deprived of oxygen during the fermentation process [29,42]. However, VHb can enhance the oxygen uptake efficiency and ATP synthesis rate in yeast cells [43,44]. Thus, coexpression of VHb could effectively improve the ability of recombinant strains to utilize oxygen under a lower or higher DO concentration, which would result in higher production of YlLip2. In addition, more than half of the energy consumption is used for aeration and stirring in high-density fermentation, which greatly increases manufacturing costs [30]. It can be seen that coexpression of VHb is of great significance to alleviate the limitation of DO and manufacturing costs in industrial-scale production.

## 4. Materials and Methods 

### 4.1. Strains, Plasmids, and Media

*Escherichia coli* Top10, *P. pastoris* GS115, pPICZA, and pPIC3.5K (Invitrogen, Carlsbad, CA, USA) were used for gene manipulation and expression. PrimeStar HS DNA polymerase, T4 DNA ligase, and pMD19-T vector were purchased from TaKaRa Biotechnology Co., Ltd (Dalian, China). All strains and vectors are listed in Appendix A. 

The *P. pastoris* strains were grown in YPD medium (20 g/L glucose, 20 g/L tryptone, and 10 g/L yeast extract), MD medium (20 g/L glucose, 13.4 g/L yeast nitrogen base with no amino acids (YNB), and 0.4 mg/L biotin), BMGY medium (10 g/L glycerol, 20 g/L tryptone, 10 g/L yeast extract, 100 mL/L potassium phosphate buffer, 13.4 g/L YNB, and 0.4 mg/L biotin), or BMMY medium (5 mL/L absolute methanol, 20 g/L tryptone, 10 g/L yeast extract, 100 mL/L potassium phosphate buffer, 13.4 g/L YNB, and 0.4 mg/L biotin). In addition, YPD-G418 medium contained 0.25 g/L geneticin (G418), and YPDS-zeocin medium contained 1 M D-sorbitol and 100 g/L zeocin. The *E. coli* strains were grown in LB medium (10 g/L sodium chloride, 5 g/L yeast extract, and 10 g/L tryptone) containing 100 μg/mL ampicillin or LLB medium (5 g/L yeast extract, 5 g/L sodium chloride, and 10 g/L tryptone) containing 25 μg/mL zeocin. For solid media, 20 g/L agar was also added. 

### 4.2. Site-Directed Mutagenesis and Vector Construction

To mutate the 244 Cys site and remove the restriction enzyme cutting site *Bgl*II in the *lip2* gene, the original *lip2* gene (Genbank accession no. AJ012632.1) without the signal peptide was used as a template to amplify four DNA fragments with the primer pairs lip2-F/Bgl-R1, Bgl-F1/Bgl-R2, Bgl-F2/244-R, and 244-F/lip2-R. The aforementioned four PCR products were ligatured via overlap extension PCR and double-digested with *Eco*RI/*Not*I. Then, the enzyme-digested fragment was linked up with *Eco*RI/*Not*I-digested pPICZαA (Δ*Sal*I) to generate pPICZαA-lip2. The plasmids pPICZαA-lip2 and pAOα-ROL [20] were double-digested with *Eco*RI/*Bam*HI to construct pAOα-lip2, a pAO815 derivative vector carrying the mutational *lip2* gene (Appendix A). All vectors and primers are listed in Appendix A. 

An efficient method was developed to construct expression vectors harboring different gene copy numbers in our previous studies [20,45]. The strategy can also be used to explore the effect of gene dosage on YlLip2 production in *P. pastoris*. The details are as follows (Appendix A): the expression cassette and part of the *his4* gene were separated from the pAOα-lip2 via double digestion with *Sal*I/*Bam*HI, and the other part of the *his4* gene, another expression cassette, the pBR322 docking platform, and the ampicillin gene were disjoined from another pAOα-lip2 vector by double digestion with *Sal*I/*Bgl*II. Then, both of the fragments were ligated and used to transform *E. coli* Top10 cells. The pAOα-2lip2 vector, containing two copies of the *lip2* gene, could be generated by cloning the intact *his4* gene with the primer pair his4-F/his4-R. In this way, the pAOα-lip2 and the pAOα-2lip2 were double-digested with *Sal*I/*Bam*HI and *Sal*I/*Bgl*II to generate pAOα-3lip2. Moreover, two pAOα-2lip2 vectors were used to construct the pAOα-4lip2 by double digestion with *Sal*I/*Bam*HI and *Sal*I/*Bgl*II. Additionally, the pAOα-nlip2 (*n* = 1–4) vectors were double-digested with *Bgl*II and *Bam*HI for further confirmation.

The vectors pPICZA-hac1 and pPICZA-vgb have been described by Jiao et al. [21] and Wang et al. [32], respectively. The pPICZA-hac1 and pPIC3.5k were double-digested with *Eco*RI/*Not*I to construct pPIC3.5k-hac1 (Appendix A). DNA sequencing for the above-described vectors was supported by TsingKe Biological Technology Co., Ltd. (Wuhan, China).

### 4.3. Transformation and Screening 

The vectors pAOα-nlip2 (*n* = 1–4) were linearized by *Sal*I and used to transform *P. pastoris* GS115 competent cells according to the recommendation of the Pichia Expression Kit Manual (Invitrogen, Carlsbad, CA, USA). Transformants were selected on MD plates and denoted as GS115/pAOα-nlip2 (*n* = 1–4). The *Sal*I-linearized pPIC3.5k-hac1 was used to transform GS115/pAOα-2lip2 #15 and was screened on YPD-G418 plates. The *Bst*XI-linearized pPICZA-vgb was separately introduced into GS115/pAOα-2lip2 #15 and GS115/2lip2-3.5Khac1 #32 and then selected on YPDS-zeocin plates. After incubation for about 72 h at 28 °C, the colonies from each recombinant strain were randomly picked on BMMY medium plates supplemented with 15 mL/L emulsified olive oil and 0.08 g/L rhodamine B. In addition, 200 μL of methanol was added daily as the inducer for recombinant strains. 

### 4.4. Shaking Flask Cultures and Optimizing Culture Parameters

Transformants producing clear and large transparent halos were chosen and placed into 5 mL of YPD medium overnight. Then, 1.5 mL cell suspension was inoculated into 50 mL of BMGY medium (initial pH 6.0) in 500 mL shaking flasks for 24 h at 28 °C. Next, yeast cells were harvested and resuspended in 500 mL shaking flasks with 50 mL of BMMY medium and then incubated for 96 h at 28 °C. In addition, 1% (*v/v*) absolute methanol was added to shaking flasks daily as the inducement for YlLip2 production. To ascertain the optimized fermentation conditions for YlLip2 expression in shaking flasks, several culture parameters, such as initial pH, incubation time, methanol concentration, culture medium volume, induction temperature, and inoculation density, were successively tested.

### 4.5. Lipase Activity and Total Protein Concentration 

The YlLip2 activity of the cell culture supernatant was measured according to a titrimetric method [8]. The reaction mixture composed of 1 mL of suitably diluted lipase solution, 5 mL of Tris-HCl buffer (50 mM, pH 8.0), and 4 mL of substrate (3 mL of 2% (*m*/*v*) polyvinyl alcohol solution emulsified with 1 mL of olive oil) was processed at 35 °C for 10 min. Then, 15 mL of cold acetone/ethanol (1:1, *v/v*) was added to the reaction system. Using phenolphthalein as an indicator, liberated fatty acid was titrated with 50 mM NaOH. One unit (U) of lipase activity could be defined as the lipase activity liberating 1 μmol of fatty acid from equivalent reaction substrate per minute. The total protein level was measured using BSA as a standard in terms of a described method [46]. All experiments were performed in triplicate.

### 4.6. Biochemical Characterization Analysis 

Early studies have indicated that YlLip2 showed higher lipase activity in the range of pH 7.5–8.5 and 35–40 °C [10,11,12]. Thus, the relative activities of the mutational YlLip2 at 40 °C in different pH (6.0–10.0) Tris-HCl buffers were measured via the above method. Analogously, the relative activities of mutational YlLip2 in pH 8.0 Tris-HCl buffer from 20 to 50 °C were determined in the same way. Furthermore, the samples were preincubated at 60, 70, and 80 °C for different times and residual activities were measured to confirm the thermostability of mutational YlLip2. All experiments were performed in triplicate.

### 4.7. Determination of Gene Copy Numbers

The copy number of target genes in transformants exhibiting higher lipase activity was determined via qPCR on the DNA Engine Opticon Monitor 2 with the MJ-Opticon 2 system (MJ Research, MA, USA). According to the product manual of SYBR Green dye (Tiangen Biotech Co., Beijing, China), qPCRs were performed in a 20 μL mixture consisting of 7.8 μL of Rnase-free water, 0.6 μL of primer pairs (10 μM), 1 μL of template, and 10 μL of 2× SuperReal PreMix Plus (Tiangen Biotech Co., Beijing, China). The amplification reactions were initiated with 15 min at 95 °C, followed by 40 cycles of 10 s at 95 °C, 20 s at 58 °C, and 30 s at 72 °C. Then, the specificity of the amplification process was analyzed by constructing melting curves. The glyceraldehyde-3-phosphate dehydrogenase gene (Genbank accession no. U62648.1, *GAP*) of *P. pastoris* was set as the endogenous reference gene and cloned by the primer pair QGAP-F/QGAP-R. Tenfold dilution series of linearized vectors harboring reference or target genes were utilized as templates to build the standard curves. The copy number of the target genes was analyzed in terms of a previously described method [47]. All samples were determined in triplicate.

### 4.8. Fed-Batch Fermentation

The recombinant strains were inoculated with 120 mL of YPD medium in a 500 mL flask for 20 h at 28 °C and transferred to a 3 L fermenter (Baoxing Co., Shanghai, China) with 1.2 L of FM22 medium [20]. There were three stages throughout the fermentation process. First, the DO value would increase suddenly and quickly when the cells exhausted all the glycerol of the FM22 medium at 29 °C. Second, about 150 mL of glycerol containing 0.3 PTM4 solution was supplied to the cells until the OD_600_ reached about 150, and carbon-source starvation was continued over 30 min to exhaust the glycerol. Third, the temperature was decreased to 27 °C and the mixed inducers (500 mL absolute methanol, 500 mL 50% D-sorbitol (*w*/*v*), and 2 mL PTM4 solution per liter) were slowly fed into the fermentation. Moreover, the fermentation broth pH was controlled at 5.5 by adding of 28% (*w/w*) ammonium hydroxide, and foam was eliminated by addition of antifoam (Dowfax DF103, Midland, MI, USA). The DO level was maintained between 20% and 50% by control of aerating, agitation, and feeding rates. Specially, the DO level was maintained lower than 20% to analyze the effects of VHb on lipase secretion in recombinant strains. The OD_600_, wet cell weight, total protein concentration, and lipase activity were monitored at set intervals throughout the fermentation cultivation. 

### 4.9. Biomass and VHb Activity Analysis

The optical density of the recombinant strains was measured at 600 nm (UV1600 Spectrophotometer, Shanghai, China) after proper dilution with deionized water. To measure the wet cell weight, 10 mL batch samples were centrifuged in preweighed tubes at 8000× *g* for 8 min and washed twice with deionized water. All samples were determined in triplicate. Additionally, the protein activity of VHb coexpressing samples was determined by the carbon monoxide difference spectrum according to a previous report [48].

### 4.10. SDS-PAGE and Mass Spectrometry Analysis

SDS-PAGE was performed with 12% separating polyacrylamide gel at 110 V for 100 min on a vertical mini gel apparatus (Bio-Rad, Hercules, CA, USA). The protein molecular weight marker was provided by GeneDirex (Flint Place Poway, CA, USA) and the Coomassie Brilliant Blue R-250 was ordered from Takara Biotechnology Co., Ltd (Dalian, China) to stain protein bands. Mass spectrometry was offered by YanXing Biological Technology Co. (Wuhan, China) for further confirmation of the target protein band.

## 5. Conclusions

A thermostable YlLip2 mutant was successfully overexpressed in *P. pastoris* via a combined strategy. First, gene dosage and fermentation parameters were separately optimized, and then Hac1p and VHb were coexpressed with YlLip2 to increase the YlLip2 expression level. On this basis, high-density cultivation was employed to obtain higher lipase activity. These strategies resulted in a great improvement of the YlLip2 mutant expression level in *P. pastoris*, and the eventual activity of the YlLip2 mutant increased 36.5-fold (from 255 to 9080 U/mL), which provides a consolidated basis for future industrial applications.

## Figures and Tables

**Figure 1 ijms-21-00279-f001:**
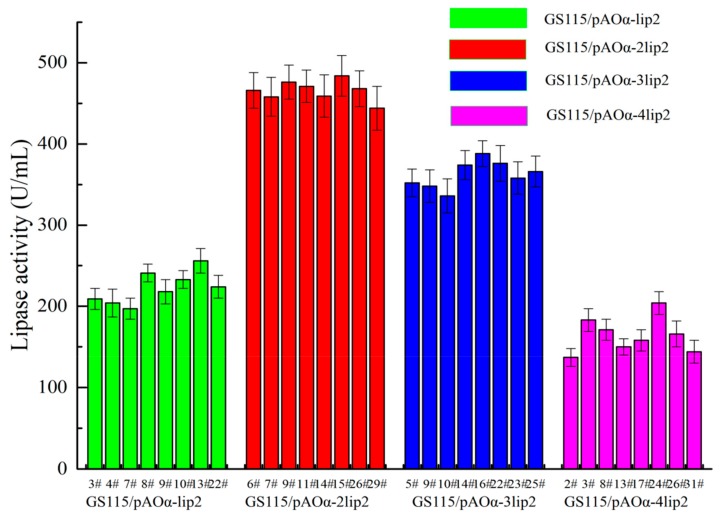
Lipase activity of recombinant strains harboring different copies of *lip2* gene expression cassettes. Data are the mean ± standard deviation of triplicate experiments.

**Figure 2 ijms-21-00279-f002:**
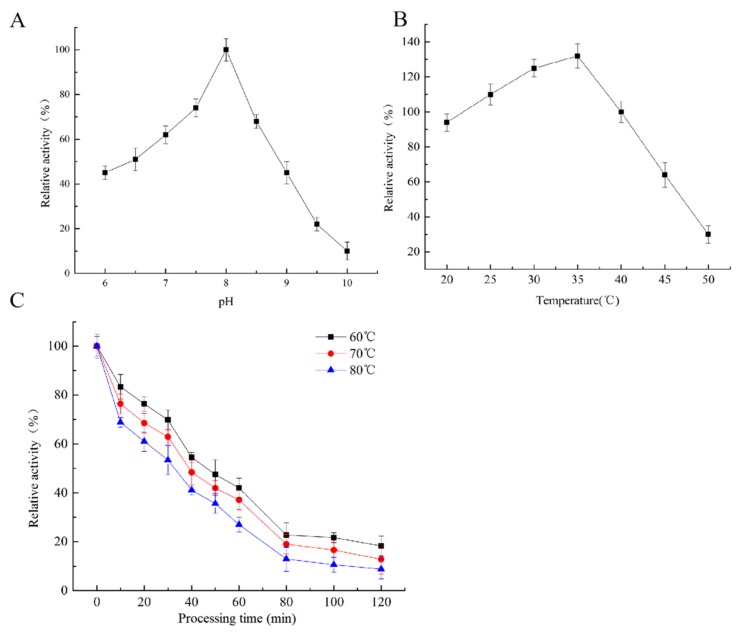
Characterization analysis for the YlLip2 mutant: (**A**) relative activity of the YlLip2 mutant in Tris-HCl buffers with different pH values; (**B**) relative activity of the YlLip2 mutant at different temperatures; and (**C**) relative activity of the YlLip2 mutant after preincubation at 60, 70 and 80 °C, respectively. Data are the mean ± standard deviation of triplicate experiments.

**Figure 3 ijms-21-00279-f003:**
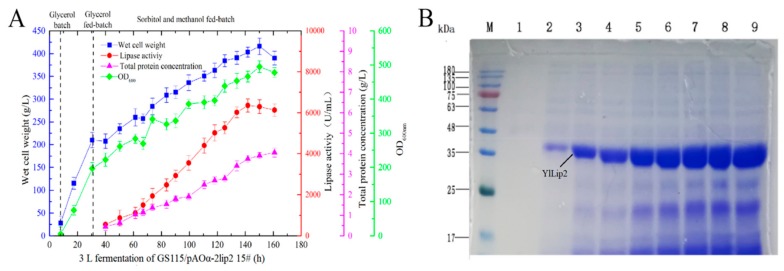
(**A**) Wet cell weight, lipase activity, total protein concentration, and optical density at 600 nm (OD_600_) during high-density fermentation of recombinant strain GS115/pAOα-2lip2 #15 in a 3 L fermenter. (**B**) SDS-PAGE analysis of GS115/pAOα-2lip2 #15 supernatants during high-cell-density fermentation in a 3 L fermenter. Lane M, molecular weight marker; lanes 1–9, culture supernatants of GS115/pAOα-2lip2 #15 at 8, 30, 50, 74, 90, 110, 125, 141, and 161 h. Data are the mean ± standard deviation of triplicate experiments.

**Figure 4 ijms-21-00279-f004:**
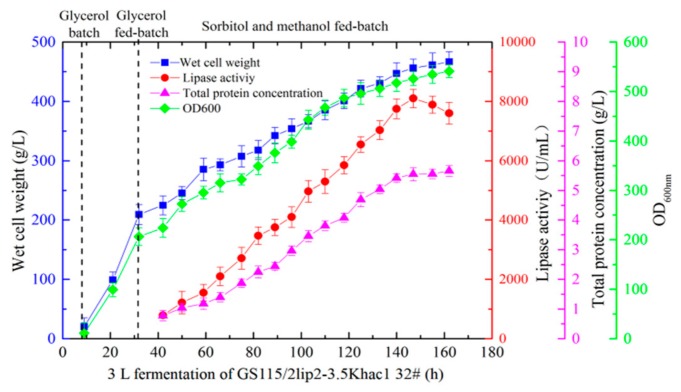
Wet cell weight, lipase activity, total protein concentration, and OD_600_ during high-density fermentation of recombinant strain GS115/2lip2-3.5Khac1 #32 in a 3 L fermenter. Data are the mean ± standard deviation of triplicate experiments.

**Figure 5 ijms-21-00279-f005:**
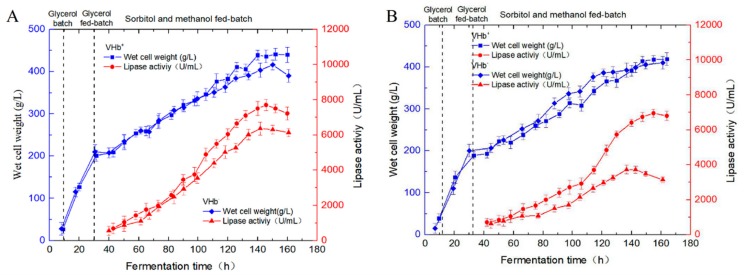
(**A**) Comparison of wet cell weight and lipase activity for GS115/pAOα-2lip2 #15 and GS115/2lip2-ZAvgb #13 in a 3 L fermenter under the higher dissolved oxygen (DO) concentration (about 50%). (**B**) Comparison of wet cell weight and lipase activity for GS115/pAOα-2lip2 #15 and GS115/2lip2-ZAvgb #13 in a 3 L fermenter under the lower DO concentration (about 10%). Data are the mean ± standard deviation of triplicate experiments.

**Figure 6 ijms-21-00279-f006:**
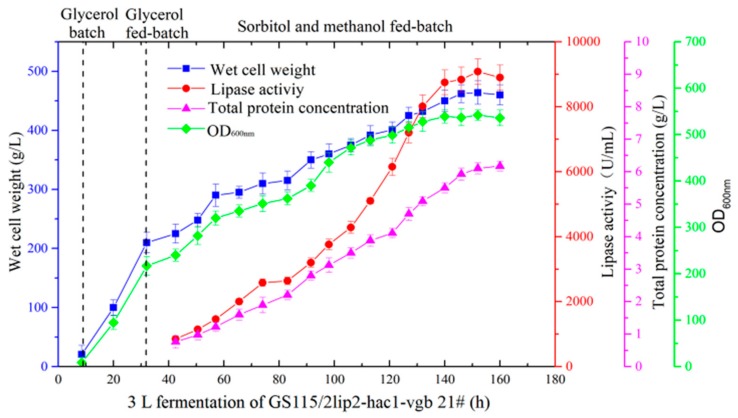
Wet cell weight, lipase activity, total protein concentration, and OD_600_ during high-density fermentation of recombinant strain GS115/2lip2-hac1-vgb #21 in a 3 L fermenter. Data are the mean ± standard deviation of triplicate experiments.

**Table 1 ijms-21-00279-t001:** qPCR analysis for recombinant strains.

Recombinant Strains	Genes Copy Number		
	*lip2*	*hac1*	*vgb*
GS115	-	1	-
GS115/pAOα-lip2 #13	1	-	-
GS115/pAOα-2lip2 #15	2	-	-
GS115/pAOα-3lip2 #16	3	-	-
GS115/pAOα-4lip2 #24	4	-	-
GS115/2lip2-3.5Khac1 #32	2	2	-
GS115/2lip2-ZAvgb #13	2	-	1
GS115/2lip2-hac1-vgb #21	2	2	1

**Table 2 ijms-21-00279-t002:** Comparison of the YlLip2 mutant expression levels in *Pichia pastoris*.

Host Strains	YlLip2 Mutants	Half-Life Time	Lipase Activity	Reference
*P. pastoris* X-33	S214 + 69 + 196 + 197 + 210	24.76 min at 70 °C	86 U/mL in shaking flask	[10]
*P. pastoris* X-33	S214 + 69 + 196 + 197 + 210 + 216	101.93 min at 70 °C	66 U/mL in shaking flask	[10]
*Y. lipolytica* JMY1212	Cys 244Ala	36 min at 75 °C	2.02 U/mL in shaking flask	[12]
*P. pastoris* GS115	Cys 244Ala	over 30 min at 80 °C	2630 U/mL in shaking flask	This study

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
