# Peer review of "High-Level Production of a Thermostable Mutant of Yarrowia lipolytica Lipase 2 in Pichia pastoris"

_ijms, 2019, doi:10.3390/ijms21010279_

Round 1

Reviewer 1 Report

The manuscript by Zhou et al. report the optimisation of a protocol for the production of Yarrowia lipolytica lipase in Pichia pastoris cells. By using Pichia pastoris instead of Yarrowia lipolytica, optimising the culture conditions, and co-expressing the mutated lip2 gene (coding for a version of the enzyme more tolerant to high temperatures compared to the wt enzyme) with the genes Hac1p and VHb, the authors were able to increase the amount of enzyme produced by more than 35 folds.

Despite the study and the relative results are of great interest, the manuscript has several problems that the authors should resolve.

Major comments

- The manuscript seems to be written by at least two people with very different writing styles, of which one is unfortunately not clear. Authors should check the whole text and ensure that the style is homogeneous and every part of the study is clear. It was really complicated to understand some part of the manuscript – please see further details in the ‘minor comments’ section.

- In the introduction, the authors clearly state that the limit in the usage of the thermostable mutated lipase (Cys244Ala) is that this mutant has very low lipase activity (2.02 U/ml). First of all, the authors should consider that the lipase activity of the wild type enzyme in the study on the Cys244Ala mutant was 2.30 U/ml, hence the reduction of activity is not as impressive as suggested by the authors of this manuscript (they report that wt enzymes can have activities up to 12500 U/ml and 42900 U/ml depending on the conditions used for the production). Secondly, authors should further stress/write more clearly why their study is important. Aiming at this, authors should expand further the limitations of the current protocols used to produce YiLip2 in Y. lipolytica; e.g. it is not clear why improving the production of the thermostable YlLip2 mutant would eliminate the problem of low lipase activity (L34-39).

- I honestly do not understand the need for characterising the mutant lipase. The effect of the Cys244Ala mutation on the activity and characteristics of the enzyme have been previously reported and described (Bordes et al. 2011, as cited by the authors). Maybe I have misunderstood the work and the used “mutant lis2” is not the Cys244Ala mutant?

- Section “2.1 Lipase activity analysis for transformants with different lip2 gene copies”. Further details should be provided. To my understanding, the authors tested here the lipase activities of culture broth of different P. pastoris strains transformed with from one to 4 copies of the lip2 gene. I believe that a title “Lipase activity analysis for transformants with different copies of the lip2 gene” would be clearer and help the reader immediately understanding the aim of the section. In addition, the authors should briefly describe the way they generated the plasmid with multiple copies of the lip2 gene before presenting the results to help to understand the results. As it is, the text is very obscure and only after collecting the information (which were spread all over the materials and methods section and in the supplementary material file) I could understand the reported data. For instance, it would be useful to report right at the beginning of the results section the molecular map of the plasmid which is currently shown in Figure S8.

Minor comments

L13: change “lower” with “low”;

L18: I guess it should be ‘co-expressed’ instead of ‘co-expression’;

L30-32: I suggest to eliminate the initial part of the sentence (“So far, there are many reports on production of YlLip2,”) to make it clearer;

L35: please clarify the sentence. I guess it should be “were devoted”?

L91: please correct “increased” with “increase”;

L92: how your method was more “controllable and accurate” compared to the cited approach?

L120-122: please rephrase the sentence, it is not clear;

L137: was the lip2 copy number higher or lower in this study compared to previous researches? I thought it was higher, and this is why the authors need to ‘help’ the protein folding by co-expressing the Hac1 gene… also, please clarify the sentence at lines 139-140 and clearly state that the recombinant strain GS115/2lip2-3.5Khac1 presented two separate vectors (one for lip2 and one for Hac1).

L142: the authors previously reported that high-density fermentation improved the production of the lipase mutant compared to shaking flask cultures (L120-122). Yet, in following tests/optimisations they grow the yeasts in shaking flasks (both in L142 and LL159). This should be explained;

L147: please explain why the additional copy of the hac1 gene is thought to be inserted into the strain genome. Shouldn’t it be in the expression plasmid? Same question for the VHb gene at lines 166 and 201.

Figure 5: please be consistent with the use of colors. Red and blue are used to show wet cell weight and lipase activity, respectively, as shown by the color of the axis. Yet, for the points of the two plots of this figure, red and blue were used to show the profiles of cells expressing or not VHb. Authors could use different point shapes to indicate the presence/absence of VHb (e.g. squares for VHb+ cells, triangles for VHb- cells).

L193-194: please change to “Thus, aiming at further increasing the level of expression of the mutant lipase, we co-expressed both Hac1p and VHb with YlLip2.”

L162: either typical or characteristics;

L299: please correct “constructe” with “construct”;

Please note that the plural of data is data, not “dates”. Please correct accordingly legends of Figures S2, S4, S5, S7, and everywhere else in the manuscript.

Figure S2: which were the basal conditions used to test the impact of changes in the 6 parameters shown in this figure? e.g. to test the effect of pH, which methanol induction time, methanol inducement % (whatever it means), culture medium volume, temperature, and inoculum % were used?

Figure S4: please correct the title of the vertical axis (change “activit” with “activity”);

Author Response

Response to Reviewer 1 Comments

Dear Editor and Reviewer 1:

We deeply appreciate the time and effort you have spent in reviewing our manuscript entitled “High-level Production of a Thermostable Mutant of Yarrowia lipolytica Lipase 2 in Pichia pastoris (ID: IJMS-673868)”. We have carefully considered all the suggestions and comments, and made corresponding corrections via the "Track Changes" function in the revised manuscript. We do hope that the revisions will make our manuscript more acceptable for its publication. Herein, we addressed in details all the revisions according to the reviewers’ comments below:

Reviewer #1:

The manuscript by Zhou et al. reports the optimization of a protocol for the production of Yarrowia lipolytica lipase in Pichia pastoris cells. By using P. pastoris instead of Y. lipolytica, optimizing the culture conditions, and co-expressing the mutated lip2 gene (coding for a version of the enzyme more tolerant to high temperatures compared to the wild-type enzyme) with the genes Hac1p and VHb, the authors were able to increase the amount of enzyme produced by more than 35 folds.

Despite the study and the relative results are of great interest, the manuscript has several problems that the authors should resolve.

Major comments:

Point 1: The manuscript seems to be written by at least two people with very different writing styles, of which one is unfortunately not clear. Authors should check the whole text and ensure that the style is homogeneous and every part of the study is clear. It was really complicated to understand some part of the manuscript-please see further details in the ‘minor comments’ section.

Response 1: Thanks for the reviewer’s valuable comments. We are very sorry for our poor English writing. The English language and style have been carefully revised by a native speaker, and we think it is much better now.

Point 2: In the introduction, the authors clearly state that the limit in the usage of the thermostable mutated lipase (Cys244Ala) is that this mutant has very low lipase activity (2.02 U/ml). First of all, the authors should consider that the lipase activity of the wild type enzyme in the study on the Cys244Ala mutant was 2.30 U/ml, hence the reduction of activity is not as impressive as suggested by the authors of this manuscript (they report that wild-type enzymes can have activities up to 12500 U/ml and 42900 U/ml depending on the conditions used for the production). Secondly, authors should further stress/write more clearly why their study is important. Aiming at this, authors should expand further the limitations of the current protocols used to produce YlLip2 in Y. lipolytica; e.g. it is not clear why improving the production of the thermostable YlLip2 mutant would eliminate the problem of low lipase activity (L34-39).

Response 2: Thanks for the reviewer’s good comments. The related part has been rewritten according to the reviewer’s suggestions in the revised manuscript. Please see Lines 39-43 in the revised manuscript. There, we will further explain why we manage to overexpress Cys244Ala-mutant YlLip2? As you know, we have succeeded in overexpressing wild-type YlLip2 whose activity can reach up to 12,500 U/ml and 42,900 U/ml depending on the conditions used for the production. But the wild-type YlLip2 would be inactivated when catalyzing reactions at higher temperatures, which greatly limits its application in industries. Whereas, Cys244Ala-mutant YlLip2 can bear high temperature over 80°C, indicating it is a potential mutant of prosperous prospect in industrial applications. While before we started this work, the activity of Cys244Ala-mutant YlLip2 was very low (2.02 U/ml) in Y. lipolytica, which was 12% lower than the wild-type YlLip2 observed in shaking flask (2.3 U/mL). So, the activity of the Cys244Ala-mutant YlLip2 remains nearly 90% of the wild-type lipase. If we can overexpress it via gene manipulation technology in some hosts, it can be available to the demand of industry.

  Why did we use P. pastoris as the host, not Y. lipolytica? The reasons are listed below: compared with Y. lipolytica, there are various molecular and genetic tools in P. pastoris, the target gene can be easily integrated into the P. pastoris genome, and the downstream separation and purification of heterologous protein are also very convenient. The gene engineered strain of P. pastoris can achieve high-density fermentation for large-scale production. Therefore, we used P. pastoris instead of Y. lipolytica as the host strain for overexpression of Cys244Ala-mutant YlLip2 in this work.

Point 3: I honestly do not understand the need for characterizing the mutant lipase. The effect of the Cys244Ala mutation on the activity and characteristics of the enzyme have been previously reported and described (Bordes et al. 2011, as cited by the authors). Maybe I have misunderstood the work and the used “mutant lip2” is not the Cys244Ala mutant?

Response 3: Thanks for the reviewer’s good comments. We are very sorry for the confusion.

  The research of Bordes et al. (2011) [1] only reported the effect of the Cys244Ala mutation on thermostability of YlLip2 in Y. lipolytica, and there may exist different characteristics between the mutant lipase produced by Y. lipolytica and that by P. pastoris. Thus, the optimum pH, the optimum temperature and the thermostability were further determined in this study. In addition, all “mutant” used in this work is the Cys244Ala mutant.

Point 4: Section “2.1 Lipase activity analysis for transformants with different lip2 gene copies”. Further details should be provided. To my understanding, the authors tested here the lipase activities of culture broth of different P. pastoris strains transformed with from one to 4 copies of the lip2 gene. I believe that a title “Lipase activity analysis for transformants with different copies of the lip2 gene” would be clearer and help the reader immediately understanding the aim of the section. In addition, the authors should briefly describe the way they generated the plasmid with multiple copies of the lip2 gene before presenting the results to help to understand the results. As it is, the text is very obscure and only after collecting the information (which was spread all over the materials and methods section and in the supplementary material file) I could understand the reported data. For instance, it would be useful to report right at the beginning of the results section the molecular map of the plasmid which is currently shown in Figure S8.

Response 4: Thanks for the reviewer’s valuable suggestions. We have changed the title “Lipase activity analysis for transformants with different lip2 gene copies” with “Lipase activity analysis for transformants with different copies of the lip2 gene”. Please see Lines 83-84 in the revised manuscript.

  Moreover, we have described the molecular map of the plasmids at the beginning of the results section. Please see Lines 85-86 in the revised manuscript.

Minor comments:

Point 5: L13: change “lower” with “low”;

Response 5: Thanks for the reviewer’s suggestion. We have changed “lower” with “low”. Please see Line 13 in the revised manuscript.

Point 6: L18: I guess it should be ‘co-expressed’ instead of ‘co-expression’;

Response 6: Thanks for the reviewer’s suggestion. We are very sorry for the error. We have made corresponding revision. Please see Line 18 in the revised manuscript.

Point 7: L30-32: I suggest to eliminate the initial part of the sentence (“So far, there are many reports on production of YlLip2,”) to make it clearer;

Response 7: Thanks for the reviewer’s suggestion. “So far, there are many reports on production of YlLip2,” has been eliminated. Please see Lines 31-32 in the revised manuscript.

Point 8: L35: please clarify the sentence. I guess it should be “were devoted”?

Response 8: Thanks for the reviewer’s suggestion. We have changed “were devoting” into “were devoted”. Please see Line 36 in the revised manuscript.

Point 9: L91: please correct “increased” with “increase”;

Response 9: Thanks for the reviewer’s suggestion. We are very sorry for the error. We have made corresponding revision. Please see Line 97 in the revised manuscript.

Point 10: L92: how your method was more “controllable and accurate” compared to the cited approach?

Response 10: Thanks for the reviewer’s good comment. When the cited approach (posttransformational vector amplification, PTVA) is used to explore the effect of gene dosage in P. pastoris, the amplification progress of target gene is unpredictable and uncontrollable. The exact copy number of target gene in recombinant strains cannot be known without determination. It cannot be confirmed that the obtained recombinant strains carry the optimal gene copy number. Moreover, it is money-costing and time-consuming to screen various multi-copy strains. In this study, only 32 colonies with different copies of lip2 gene were inoculated into BMGY medium and then the optimal gene dosage for YlLip2 production in P. pastoris was screened. In addition, all transformants harboured the exactly same lip2 gene copy numbers as the corresponding plasmids did. Therefore, compared with the cited approach, the method in this study was more “controllable and accurate”.

Point 11: L120-122: please rephrase the sentence, it is not clear;

Response 11: Thanks for the reviewer’s suggestion. We have rephrased the sentence and made it readable. Please see Lines 128-132 in the revised manuscript.

Point 12: L137: was the lip2 copy number higher or lower in this study compared to previous researches? I thought it was higher, and this is why the authors need to ‘help’ the protein folding by co-expressing the hac1 gene… also, please clarify the sentence at lines 139-140 and clearly state that the recombinant strain GS115/2lip2-3.5Khac1 presented two separate vectors (one for lip2 and one for Hac1).

Response 12: Thanks for the reviewer’s good comments. Wang et al. [2] reported a recombinant strain (whose activity reached 33,000 U/mL in 10-L fermenter) with 7 copies of wild-type lip2 gene, and Yu et al. [3] obtained a transformants (whose activity reached 42,900 U/mL in 5-L fermenter) with 3.4 copies of wild-type lip2 gene. In comparison to the previous researches, the optimal copy number of the mutated lip2 gene was lower. We suspected that the YlLip2 mutant would be more difficult to fold and translate than the wild-type YlLip2. So, co-express Hac1p to ‘help’ the protein folding so as to enhance the production of the YlLip2 mutant (or wild-type YlLip2) in P. pastoris.

  The two separate vectors (pAOα-2lip2 and pPIC3.5k-hac1) were successively linearized and transformed into the host strain GS115 to obtain recombinant strain GS115/2lip2-3.5Khac1. The two vectors were both inserted into the yeast genome by homologous recombination, and there were no episomal vectors in GS115/2lip2-3.5Khac1.

Point 13: L142: the authors previously reported that high-density fermentation improved the production of the lipase mutant compared to shaking flask cultures (L120-122). Yet, in following tests/optimizations they grow the yeasts in shaking flasks (both in L142 and LL159). This should be explained;

Response 13: Thanks for the reviewer’s good comments. Generally speaking, shaking flask culture is convenient for screening recombinant strains, while high-density fermentation is suitable for large-scale protein production. Thus, the shaking flask culture parameters were optimized for the subsequently screening recombinant strains. Meanwhile, the secretion level of YILip2 in P. pastoris was improved under the optimized culture parameters.

Point 14: L147: please explain why the additional copy of the hac1 gene is thought to be inserted into the strain genome. Shouldn’t it be in the expression plasmid? Same question for the vgb gene at lines 166 and 201.

Response 14: Thanks for the reviewer’s good comments. We are very sorry for the confusion. P. pastoris has no stable episomal plasmids, foreign gene vectors are readily inserted into its genome by homologous recombination. In this study, both the hac1 (or vgb) gene and other part of the expression plasmid pPIC3.5k-hac1 (or pPICZA-vgb) were inserted into yeast genome. In addition, there is one original hac1 gene in GS115 genome. Thus, single copy of additional hac1 gene was inserted into the GS115/2lip2-3.5Khac1 32# genome and only one copy of vgb gene was inserted into the GS115/2lip2-ZAvgb 13# genome.

Point 15: Figure 5: please be consistent with the use of colors. Red and blue are used to show wet cell weight and lipase activity, respectively, as shown by the color of the axis. Yet, for the points of the two plots of this figure, red and blue were used to show the profiles of cells expressing or not VHb. Authors could use different point shapes to indicate the presence/absence of VHb (e.g. squares for VHb+ cells, triangles for VHb- cells).

Response 15: Thanks for the reviewer’s suggestions. We have made corresponding revisions. Please see Figure 5 in the revised manuscript.

Point 16: L162: either typical or characteristics;

Response 16: Thanks for the reviewer’s suggestion. We have changed “a typical characteristic absorption peak” into “a characteristic absorption peak”. Please see Line 173 in the revised manuscript.

Point 17: L193-194: please change to “Thus, aiming at further increasing the level of expression of the mutant lipase, we co-expressed both Hac1p and VHb with YlLip2.”

Response 17: Thanks for the reviewer’s suggestion. We have made corresponding revisions. Please see Lines 204-206 in the revised manuscript.

Point 18: L299: please correct “constructe” with “construct”;

Response 18: Thanks for the reviewer’s suggestion. We are very sorry for the error. We have made corresponding revision. Please see Line 242 in the revised manuscript.

Point 19: Please note that the plural of data is data, not “dates”. Please correct accordingly legends of Figures S2, S4, S5, S7, and everywhere else in the manuscript.

Response 19: Thanks for the reviewer’s suggestions. We are very sorry for the errors. We have made corresponding revisions. Please see Figures 1, 2, 3, 4, 5, 6, S4, S6, S7 and S9 in the revised manuscript.

Point 20: Figure S2: which were the basal conditions used to test the impact of changes in the 6 parameters shown in this figure? e.g. to test the effect of pH, which methanol induction time, methanol inducement % (whatever it means), culture medium volume, temperature, and inoculum % were used?

Response 20: Thanks for the reviewer’s comments. The basal conditions used for test the impact of changes in the 6 parameters shown in Figure S2 (corresponding Figure S4 in the revised supplementary file) has been added in “3.4. Shaking flask cultures and optimizing culture parameters” in the revised manuscript (Please see Lines 273-275). Specifically, the basal conditions were initial pH 6.0, addition of 1% (v/v) absolute methanol daily, initial growth medium volume 50 mL, inoculation density 3% (v/v), and fermentation for 96 h at 28 °C.

Point 21: Figure S4: please correct the title of the vertical axis (change “activit” with “activity”);

Response 21: Thanks for the reviewer’s suggestion. We are very sorry for the error. We have change “activit” with “activity” in Figure S6 of the revised manuscript.

In the end, we are very grateful for the reviewer and the editor for their kindness!

References

Bordes, F.; Tarquis, L.; Nicaud, J.; Marty, A. Isolation of a thermostable variant of Lip2 lipase from Yarrowia lipolytica by directed evolution and deeper insight into the denaturation mechanisms involved. J. Biotechnol. 2011, 156, (2), 117-124. https://doi.org/10.1016/j.jbiotec.2011.06.035. Wang, X.; Sun, Y.; Shen, X.; Ke, F.; Zhao, H.; Liu, Y.; Xu, L.; Yan, Y. Intracellular expression of Vitreoscilla hemoglobin improves production of Yarrowia lipolytica lipase LIP2 in a recombinant Pichia pastoris. Enzyme Microb. Tech. 2012, 50, (1), 22-28. https://doi.org/10.1016/j.enzmictec.2011.09.003. 3. Yu M.; Wen S.; Tan T. Enhancing production of Yarrowia lipolytica lipase Lip2 in Pichia pastoris. Life Sci. 2010, 10(5): 458-464. https://doi.org/10.1002/elsc.200900102.

Reviewer 2 Report

Review of IJMS-673868 High-level Production of a Thermostable Mutant of 3 Yarrowia lipolytica Lipase 2 in Pichia pastoris
authors Qinghua Zhou, Zhixin Su, Liangcheng Jiao, Yao Wang, Kaixin Yang, Wenjuan Li,Yunjun Yan

This well written manuscript describes a combination of methods to incease the efficiency of production of an enzyme important for industrial applications, Lipase 2 from the yeast Yarrowia lipolytica.
The manuscript suffers from some language problems, I have listed below some of them, but these do not prevent understanding. As it stands, the manuscript is a technical recipee for producing an enzyme, it is not a scientific contribution.
The technique is useful and important, but should be published in a biotechnological journal. This is a decision for the editor of IJMS, but I remind here the S stand for science.
This situation could be changed if the authors were to explain why the specific combination of conditions improve production. Put in the context of other such efforts, for other enzymes, and compared with the conditions improving efficiencies for other enzymes.
For example, does teperature always have the same effects/optima? Why? is the optimum associated with enzyme size, or some other property of the enzyme?
The same kind of questions for shaking etc could be asked and discussed comparatively with existing literature. Common principles adn explanations, included for differences between enzymes, could be suggested.
These developments, in relation to other similar cases, would give a frame that is beyond technical to this effort and contribute to develop a more general theory (meaning scientific frame) for this type of experiments, beyond improving the means of production.

Introduction 41 lines
Result s 127 lines->there is no discussion
M+M 116 lines
Conclusions 7 lines
line
91 increased->increase
Figure 1->what are the different colours for?
94 Fig 1 legend. what Dates? do you mean data?
101 which can meet THE majority of ...
106 Fig 2 legend. what Dates? do you mean data?
125 presented->present This->It
134 Fig 3 legend. what Dates? do you mean data?
137 was lower than the->for previous researches->analyses or experiments
153 Fig 4 legend. what Dates? do you mean data?
186 Fig 5 legend. what Dates? do you mean data?
189 Fig 6 legend. what Dates? do you mean data?
229 constructe->construct
276 experimentS
285 experimentS was->were
287 determinated ->determined
312 fewer->lower
325 maker->do you mean marker?
328 for further confirmation of THE target-protein band.
332 co-expression->co-expressed

Author Response

Response to Reviewer 2 Comments

Dear Editor and Reviewer 2:

We deeply appreciate the time and effort you have spent in reviewing our manuscript entitled “High-level Production of a Thermostable Mutant of Yarrowia lipolytica Lipase 2 in Pichia pastoris (ID: IJMS-673868)”. We have carefully considered all the suggestions and comments, and made corresponding corrections via the "Track Changes" function in the revised manuscript. We do hope that the revisions will make our manuscript more acceptable for its publication. Herein, we addressed in details all the revisions according to the reviewers’ comments below:

Reviewer #2:

This well written manuscript describes a combination of methods to increase the efficiency of production of an enzyme important for industrial applications, Lipase 2 from the yeast Yarrowia lipolytica.

The manuscript suffers from some language problems, I have listed below some of them, but these do not prevent understanding. As it stands, the manuscript is a technical recipe for producing an enzyme, it is not a scientific contribution.

The technique is useful and important, but should be published in a biotechnological journal. This is a decision for the editor of IJMS, but I remind here the S stand for science.

Point 1: This situation could be changed if the authors were to explain why the specific combination of conditions improves production. Put in the context of other such efforts, for other enzymes, and compared with the conditions improving efficiencies for other enzymes.

For example, does temperature always have the same effects/optima? Why? Is the optimum associated with enzyme size, or some other property of the enzyme?

The same kind of questions for shaking etc. could be asked and discussed comparatively with existing literature. Common principles and explanations, included for differences between enzymes, could be suggested.

These developments, in relation to other similar cases, would give a frame that is beyond technical to this effort and contribute to develop a more general theory (meaning scientific frame) for this type of experiments, beyond improving the means of production.

Response 1: Thanks for the reviewer’s valuable comments. The optimized culture parameters for protein production in recombinant strains are affected by many factors. Thereinto, the host strain and targeted protein play decisive roles. In addition, a single parameter may have several effects on protein production in recombinant strains. For example, the cell growth and metabolism are greatly influenced by temperature, and the specific mechanism of action is too complicated to describe in a few sentences. Thus, the optimized combination of conditions for different strain or protein was usually divergent. A common principle is suitable for the process of optimization: all parameters were changed for protein production.

Specifically, temperature does not always have the same effects in different proteins or strains. The optimized culture parameters were associated with characterization of protein including enzyme size. Shaking also affects protein production but 200 rpm is appropriate for most proteins.

In this study, the shaking flask culture parameters were successively tested for subsequently screening recombinant strains. Meanwhile, the secretion level of YILip2 in P. pastoris was significantly improved under the optimized culture parameters. Beside, high-density fermentation is more convenient for large-scale protein production. Therefore, we did not discuss too much in this work.

Point 2:  Introduction 41 lines

Results 127 lines->there is no discussion

M+M 116 lines

Conclusions 7 lines

Response 2: Thanks for the reviewer’s comment. “Discussion” was combined with “Results”, Please see “Results and discussion” in the revised manuscript.

Point 3: line 91 increased->increase

Response 3: Thanks for the reviewer’s suggestion. We are very sorry for the error. We have made corresponding revision. Please see Line 97 in the revised manuscript.

Point 4: Figure 1->what are the different colours for?

Response 4: Thanks for the reviewer’s comment. Four colours were used to show the lipase activity of recombinant strains harboring different copies of lip2 gene expression cassettes in Figure 1. To avoid confusing of using different colours, we have revised the Figure 1 in the revised manuscript.

Point 5: line 94 Fig 1 legend. What Dates? Do you mean data?

Response 5: Thanks for the reviewer’s comments. We are very sorry for the error. We have change “Dates” into “Data” in Figure 1 legend. Please see Line 100 in the revised manuscript.

Point 6: line 101 which can meet THE majority of ...

Response 6: Thanks for the reviewer’s suggestion. We have rephrased the sentence and made it readable. Please see Lines 107-109 in the revised manuscript.

Point 7: line 106 Fig 2 legend. What Dates? Do you mean data?

Response 7: Thanks for the reviewer’s comments. We are very sorry for the error. We have change “Dates” into “Data” in Figure 2 legend. Please see Line 113 in the revised manuscript.

Point 8: line 125 presented->present This->It

Response 8: Thanks for the reviewer’s suggestions. We have made corresponding revisions. Please see Line 135 in the revised manuscript.

Point 9: line 134 Fig 3 legend. What Dates? Do you mean data?

Response 9: Thanks for the reviewer’s comments. We are very sorry for the error. We have change “Dates” into “Data” in Figure 3 legend. Please see Line 144 in the revised manuscript.

Point 10: line 137 was lower than the->for previous researches->analyses or experiments

Response 10: Thanks for the reviewer’s suggestions. We have made corresponding revisions. Please see Lines 147-148 in the revised manuscript.

Point 11: line 153 Fig 4 legend. What Dates? Do you mean data?

Response 11: Thanks for the reviewer’s comments. We are very sorry for the error. We have change “Dates” into “Data” in Figure 4 legend. Please see Line 164 in the revised manuscript.

Point 12: line 186 Fig 5 legend. What Dates? Do you mean data?

Response 12: Thanks for the reviewer’s comments. We are very sorry for the error. We have change “Dates” into “Data” in Figure 5 legend. Please see Line 197 in the revised manuscript.

Point 13: line 189 Fig 6 legend. What Dates? Do you mean data?

Response 13: Thanks for the reviewer’s comments. We are very sorry for the error. We have change “Dates” into “Data” in Figure 6 legend. Please see Line 200 in the revised manuscript.

Point 14: line 229 constructe->construct

Response 14: Thanks for the reviewer’s suggestion. We are very sorry for the error. We have made corresponding revision. Please see Line 243 in the revised manuscript.

Point 15: line 276 experimentS

Response 15: Thanks for the reviewer’s suggestion. We are very sorry for the error. We have change “experiment” into “experiments”. Please see Line 291 in the revised manuscript.

Point 16: line 285 experimentS was->were

Response 16: Thanks for the reviewer’s suggestions. We are very sorry for the errors. We have change “experiment was” into “experiments were”. Please see Line 300 in the revised manuscript.

Point 17: line 287 determinated ->determined

Response 17: Thanks for the reviewer’s suggestion. We are very sorry for the errors. We have change “determinated into “determined”. Please see Line 303 in the revised manuscript.

Point 18: line 312 fewer->lower

Response 18: Thanks for the reviewer’s suggestion. We have made corresponding revisions. Please see Line 327 in the revised manuscript.

Point 19: line 325 maker->do you mean marker?

Response 19: Thanks for the reviewer’s comment. We are very sorry for the error. We have change “maker” into “marker”. Please see Line 340 in the revised manuscript.

Point 20: line 328 for further confirmation of THE target-protein band.

Response 20: Thanks for the reviewer’s suggestion. We have made corresponding revisions. Please see Line 343 in the revised manuscript.

Point 21: line 332 co-expression->co-expressed

Response 21: Thanks for the reviewer’s suggestion. We are very sorry for the error. We have made corresponding revision. Please see Line 347 in the revised manuscript.

In the end, we are very grateful for the reviewer and the editor for their kindness!

Round 2

Reviewer 2 Report

The main point of my comments, that results must be discussed within a general, scientific frame, is not adressed by this new version, which is from my point of view only slightly improved from a language point of view (see some edits below).
As a discussion, the author reply to me about effects of shaking and temperature would be a fair start and some similar text, expanded, and with references, could be the missing discussion in this context. The answer that results and discussion are mixed is inadequate, because a more general discussion is missing. See for example my comment (below) for line 113.
In that part, you need to discuss why some enzymes react in one way to temperature, and others in another way. Same about shaking etc. You need to give references to other examples.

line
12 As a promising biocatalysts->biocatalyst
32 The YlLip2 activity was reported as 12,500 U/mL in A 7.5 L bioreactor in
Pichia pastoris X33[8]
99, 111, 141, 161, 195, 197 DATA ARE
102 was conducted FIRST
113 As known->this needs references that say from where this is known
133 itcould->it could
144 than in previous experiments (delete that)

Author Response

Response to Reviewer 2 Comments

Dear Editor and Reviewer 2:

We deeply appreciate the time and effort you have spent in reviewing our manuscript entitled “High-level Production of a Thermostable Mutant of Yarrowia lipolytica Lipase 2 in Pichia pastoris (ID: IJMS-673868)”. We have carefully considered all the suggestions and comments, and made corresponding corrections via the "Track Changes" function in the revised manuscript. We do hope that the revisions will make our manuscript more acceptable for its publication. Herein, we addressed in details all the revisions according to the reviewer’ comments below:

Reviewer #2:

Point 1:The main point of my comments, that results must be discussed within a general, scientific frame, is not addressed by this new version, which is from my point of view only slightly improved from a language point of view (see some edits below).

As a discussion, the author reply to me about effects of shaking and temperature would be a fair start and some similar text, expanded, and with references, could be the missing discussion in this context. The answer that results and discussion are mixed is inadequate, because a more general discussion is missing. See for example my comment (below) for line 113.

In that part, you need to discuss why some enzymes react in one way to temperature, and others in another way. Same about shaking etc. You need to give references to other examples.

Response 1: Thanks for the reviewer’s valuable comments. We have separated “Results and discussion”, and written a separate “Discussion” section in the revised manuscript. In Discussion, we compared our research with other previous studies about optimizing culture conditions for protein expression in P. pastoris (Please see Table S1 in supplementary file). Then, we supplemented the discussion on shaking flask culture parameters and cited corresponding references. Please see Lines 221-234 in the revised manuscript.

Table S1 The optimized culture parameters for protein expression in P. pastoris.

Proteins

Temperature

(°C)

Initial pH

Methanol concentration

(v/v)

Culture medium volume (ml)

Inoculation density (v/v)

Incubation time (h)

References

YlLip2 mutant

Wild-type YlLip2

Rhizopus oryzae lipase

Pycnoporus sanguineus laccase

Human μ-opioid receptor

G-protein-coupled receptors

22

25

27

30

20

20

7.0

6.5

7.0

6.5

-

7.0

1%

1%

1.2%

0.5%

0.5%

0.5%

30

-

20

-

-

-

3%

-

4%

-

-

-

144

-

120

168

-

-

This study

[1]

[2]

[3]

[4]

[5]

In the above-mentioned studies, target proteins were secreted as an extracellular protein of the recombinant strains. The culture temperature influenced recombinant strains on cell growth and metabolism, thus, the protein expression levels were different in different culture temperatures [1-5]. For example, high temperature is beneficial for cell growth, but not for protein production because high temperature inhibits the activity of the AOX1 promoter [6] (Please see Lines 225-226 in the revised manuscript). Thus, we think it is recombinant strain rather than enzyme reacts to temperatures.

We are sorry, up to now, that there are fewer studies on the influence of shaking on protein expression in P. pastoris. Actually, the main object of shaking is to aerate the culture medium to keep cell growth [7]. It is known to all that 200 rpm is sufficient. So, we have no related experiments. Please understand we did not discuss shaking in this study.

Point 2: line 12 As a promising biocatalysts->biocatalyst

Response 2: Thanks for the reviewer’s suggestion. We are very sorry for the error. We have change “biocatalysts” into “biocatalyst”. Please see Line 12 in the revised manuscript.

Point 3: line 32 The YlLip2 activity was reported as 12,500 U/mL in A 7.5 L bioreactor in

Pichia pastoris X33

Response 3: Thanks for the reviewer’s suggestion. We have made corresponding revision. Please see Line 32 in the revised manuscript.

Point 4: line 99, 111, 141, 161, 195, 197 DATA ARE

Response 4: Thanks for the reviewer’s suggestion. We are very sorry for the errors. We have change “Date is” into “Data are”. Please see Lines 97, 109, 138, 157, 193 and 195 in the revised manuscript.

Point 5: line 102 was conducted FIRST

Response 5: Thanks for the reviewer’s suggestion. We have change “was first conducted” into “was conducted first”. Please see Line 100 in the revised manuscript.

Point 6: line 113 As known->this needs references that say from where this is known

Response 6: Thanks for the reviewer’s suggestion. We have cited corresponding references. Please see Line 112 in the revised manuscript.

Point 7: line 133 itcould->it could

Response 7: Thanks for the reviewer’s suggestion. We have change “itcould” into “it could”. Please see Line 129 in the revised manuscript.

Point 8: line 144 than in previous experiments (delete that)

Response 8: Thanks for the reviewer’s suggestion. We have made corresponding revision. Please see Line 141 in the revised manuscript.

In the end, we are very grateful to the reviewer and the editor for their kindness!

References

Wang, X.; Shen, X.; Zhao, H.; Sun, Y.; Liu, T.; Liu, Y.; Xu, L.; Yan, Y. Combined strategies for the improvement of heterologous expression of a His-tagged Yarrowia lipolytica lipase Lip2 in Pichia pastoris. Afr. J. Biotechnol. 2011, 10(80), 18503-18512. Jiao, L.; Zhou, Q.; Su, Z.; Xu, L.; Yan, Y. High-level extracellular production of Rhizopus oryzae lipase in Pichia pastoris via a strategy combining optimization of gene-copy number with co-expression of ERAD-related proteins. Protein Expr. Purif. 2018, 147, 1-12. https://doi.org/10.1016/j.pep.2018.02.005. Lu, L.; Zhao, M.; Liang, S.; Zhao, L.; Li, D.; Zhang, B. Production and synthetic dyes decolourization capacity of a recombinant laccase from Pichia pastorisJ. Appl. Microbiol2009, 107(4), 1149-1156. https://doi.org/10.1111/j.1365-2672.2009.04291.x. Sarramegna, V.; Demange, P.; Milon, A.; Talmont, F. Optimizing functional versus total expression of the human μ-opioid receptor in Pichia pastoris. Protein Expr. Purif. 2002, 24(2), 212-220. https://doi.org/10.1006/prep.2001.1564. Yurugi-Kobayashi, T.; Asada, H.; Shiroishi, M.; Shimamura, T.; Funamoto, S.; Katsuta, N.; Ito, K.; Sugawara, T.; Tokuda, N.; Tsujimoto, H.; Murata, T.; Nomura, N.; Haga, K.; Haga, T.; Iwata, S.; Kobayashi, T. Comparison of functional non-glycosylated GPCRs expression in Pichia pastoris. Biochem. Biophys. Res. Commun. 2009, 380(2): 271-276. https://doi.org/10.1016/j.bbrc.2009.01.053. 6. Noseda, D.; Recúpero, M.; Blasco, M.; Ortiz, G.; Galvagno, M. A. Cloning, expression and optimized production in a bioreactor of bovine chymosin B in Pichia (Komagataella) pastoris under AOX1 Protein Expr. Purif. 2013, 92(2), 235-244. https://doi.org/10.1016/j.pep.2013.08.018. Rajasekhar, E.; Edwards, M.; Wilson, S.; Street, H. Studies on the growth in culture of plant cells: XI. THE INFLUENCE OF SHAKING RATE ON THE GROWTH OF SUSPENSION CULTUEES. J. Exp. Bot. 1971, 22(1), 107-117. https://doi.org/10.1093/jxb/22.1.107.

Round 3

Reviewer 2 Report

The authors have separated discussion from results, and expanded their discussion of their results, in the sense that they attempt at explaining why/how some treatments improve production of the target protein. This  improved the understanding of this reviewer who has no background in biotechnologies. After reading this third version, I feel that my knowledge in this field advanced, this was not the case after the 1st and 2nd rounds of review. I thank the authors for their patience, in my view their effort improved the value of their communication, beyond the interest in describing a valuable and useful technique. 

Minor english problems:

avoid the word triumphant, this is inadequate in the scientific context

supposal->do you mean supposition